# Advances in the Biological Application of Force-Induced Remnant Magnetization Spectroscopy

**DOI:** 10.3390/molecules27072072

**Published:** 2022-03-23

**Authors:** Shuyu Liao, Mengxue Sun, Jinxiu Zhan, Min Xu, Li Yao

**Affiliations:** 1State Key Laboratory for Structural Chemistry of Unstable and Stable Species, Beijing National Laboratory for Molecular Sciences, Institute of Chemistry, Chinese Academy of Sciences, CAS Research/Education Center for Excellence in Molecular Sciences, Beijing 100190, China; liaoshuyu21@mails.ucas.ac.cn (S.L.); sunmengxue@iccas.ac.cn (M.S.); zhanjinxiu@iccas.ac.cn (J.Z.); xumin311@iccas.ac.cn (M.X.); 2School of Chemical Sciences, University of Chinese Academy of Sciences, Beijing 100049, China

**Keywords:** non-covalent bonds, molecular and cellular interactions, FIRMS, mechanical force, magnetic probe

## Abstract

Biomolecules participate in various physiological and pathological processes through intermolecular interactions generally driven by non-covalent forces. In the present review, the force-induced remnant magnetization spectroscopy (FIRMS) is described and illustrated as a novel method to measure non-covalent forces. During the FIRMS measurement, the molecular magnetic probes are magnetized to produce an overall magnetization signal. The dissociation under the interference of external force yields a decrease in the magnetic signal, which is recorded and collected by atomic magnetometer in a spectrum to study the biological interactions. Furthermore, the recent FIRMS development with various external mechanical forces and magnetic probes is summarized.

## 1. Introduction

The binding force generated by non-covalent interactions in organisms plays an important role in regulating and controlling the cellular and molecular behavior [1,2,3], involving the ligand–receptor bond, antibody–antigen bond, DNA/RNA binding, and cell adhesion. Specifically, the non-covalent bonds in biomolecular systems are the basis of particular interactions, for instance, enzyme reaction, DNA hybridization, and protein-virus binding [4,5,6]. Cell adhesion force has a prominent impact on tumor metastasis [7]. The broad role in the biological field suggests that techniques for precisely measuring non-covalent interactions are worth developing.

To date, the single-molecule force spectroscopy techniques have proven to be powerful tools for probing the non-covalent interaction of biomolecules; major examples are atomic force microscopy [8,9,10], optical tweezers [11,12] and magnetic tweezers [13,14,15]. In these techniques, the molecules under study are immobilized on the substrate, and the probe is modified with molecules involved in the non-covalent interaction. Under stress, the probe is restricted by non-covalent interactions and only makes a small displacement around its equilibrium position [16]. Then, the mechanical information of biomolecules is obtained by the function relationship between force and displacement. The accuracy and resolution are strongly affected by the site of the probe, and even the smallest environmental changes can affect the quality of the results, let alone the effect caused by low selectivity that leads the probe to contact the molecule in an undesirable position. The above significant drawback means that it is easy to provide widely distributed and unstable measured results [16,17], which reduces both the sensitivity and resolution. The lack of high resolution makes it difficult to resolve non-covalent bonds within 10 pN differences, such as the drugs–DNA system [18]. Meanwhile, the dynamic contact detection reflects the dynamic binding force between molecules, which means that the binding force at equilibrium state cannot be measured. Furthermore, the detection scale of just one molecule at a time confined by a narrow field of view imposes a restriction on the resolving multivalent interactions between the particle and surface, and the useful results are only obtained after a laborious and time-consuming repetition of a large number of measurements. Lastly, for other biological processes, such as ribosomal translocation, the problems from invasive measurement remain unsolved.

Recently, the force-induced remnant magnetization spectroscopy (FIRMS) is developed as an available technique to address the issues above, where a large number of events can be detected at once, with the highest resolution up to 0.5 pN [19]. The FIRMS technique mainly consists of three parts: a magnetically labeled sample, magnetometer sensor, and force (**F**) generator. The principle is illustrated in Figure 1. The labeling with magnetic probes provides both imaging and magnetization signals. Each magnetic probe has the same magnetization ability. Compared to the fluorescence and radiation imaging methods, magnetic labeling has the advantages of low toxicity, non-invasive detection, and easy operation [20,21]. The coherent-cesium atomic magnetometer based on the Faraday magnetorotation effect is one of the most sensitive apparatus for measuring magnetic fields [22,23]. The atomic magnetometer using the scanning imaging method obtains a magnetic field map with both spatial and quantity information [24]. In the FIRMS technique, the probes are attached to the samples, which are immobilized on the substrate, and magnetized by a permanent magnet producing an overall remnant magnetic signal. This initial magnetization value corresponds to the strongest signal measured by an atomic magnetometer. As the force generator exerts gradient force on the non-covalent bonds, the labeled samples gradually detach from the immobile surface undergoing Brownian relaxation. The magnetic dipoles become disoriented, which results in the decline of the magnetization signal intensity. The relationship between the magnetization signal and applied force is the main part of the FIRMS measurement (Figure 2). Due to the linear correlation between the magnetization and the number of various magnetic probes [25], the remnant magnetic signal at each mechanical force marks the quantity of undissociated magnetic probes, and the force at a sharp drop in the signal represents binding force.

This review focuses on the advances in the biological application of FIRMS, harnessing various external forces and magnetic probes, with the intention of assisting in the selection of a suitable mechanical force and probe to detect non-covalent interactions.

## 2. External Mechanical Force

In this section, we present various external forces that may be applied in FIRMS, including shaking force, centrifugal force, ultrasound radiation force, and fluid shear force. The implementation of these external mechanical forces plays different roles, such as the parameter in molecular recognition, reference for determining the force that cannot be measured directly, and characterization of cell adhesion.

### 2.1. Shaking Force

The shaking force is obtained by a vortex mixer, which can coarsely distinguish the difference between physical adsorption and non-covalent specific binding. Yao et al. characterized these two types of interactions via human CD3+ T cells binding to magnetic probes by the CD3 antibody [26]. The physisorbed probes separated from the cells at a low shaking force, while specifically bound antibody retained magnetic signals after increasing the speed by 1000 rpm. In order to resolve the binding type between the magnetic probes and cells in different conditions quantitatively, they took the derivative of the magnetization curve. Gaussian fitting was used to scale the force of the physical adsorption peak. By obtaining the spectra of the blank experiment as a reference and deducing from the location, intensity, and width of the peak, it was found that the cell-binding peak height was only 30% of that in the blank spectrum, indicating that the remaining 70% of the magnetic particles were non-covalently bound to the target cells by the characteristic CD3 binding pair (Figure 3). There is not enough shaking force to dissociate the non-covalent bonds.

### 2.2. Centrifugation Force

In comparison to the shaking force, the force created via a centrifuge can finely reach the needed values by adjusting revolution speed. For molecular systems, since the mass of magnetic probe is much heavier than molecule, the mass of the molecule can be neglected when calculating the centrifugal force, and the magnetic probe can be regarded as the sole carrier of centrifugal force. The centrifugal force applied on the magnetic probe is calculated as
*F* = (*ρ_m_* − *ρ_b_*) *V_m_ω*^2^*r*
where *ρ_m_* is the density of the magnetic probe, *ρ_b_* is the density of the buffer solution, *V**_m_* is the volume of magnetic probe, *r* is the distance of the sample to the rotating axis, and *ω* is the angular speed, which can be calculated by the following equation involving the revolution speed n (revolutions per minute, rpm), *ω* = 2nπ/60.

By employing a centrifugal force, the early assay revealed a well-defined binding force for the bonds between the mouse immunoglobulin G and the magnetically labeled α-mouse immunoglobulin G [25]. In Chen’s research, three types of interactions between the anti-CD4 antibody-conjugated magnetic beads and CD4+ T-cell surfaces were quantitatively measured for the first time [27]. The binding environment was found to have an effect on the strength of ligand–receptor bonds when comparing the interactions on both primitive cell surfaces and a functionalized surface. In addition, the centrifugal force exerted onto the cell surfaces is expected to become a quantitative parameter for discriminating multivalent interactions of ligand–receptor bonds. On this basis, recent research allowed FIRMS to resolve the single-, double-, and triple-biotin–streptavidin interactions, multivalent DNA interactions, and CXCL12–CXCR4 interactions from a macroscopic field of view [19]. De Silva et al. reported that the narrow force distribution renders the distinction of two DNA duplexes with a single-base-pair difference. The results indicate that the interaction of the two chains in DNA oligomers relies on the site of the mismatched base (Figure 4a) [28]. Yao et al. utilized the centrifugal force as a force ruler for rupturing DNA–mRNA duplexes of 13-to-18 base pairs, which allowed researchers to non-invasively obtain the information of the power stroke of the motor protein elongation factor G (EF-G) during the ribosome translocation (Figure 4b) [29]. Yin et al. determined that the quantitative measurements of the power strokes of structurally modified EF-Gs by using FIRMS and the microscope-based technique revealed the correlation between power stroke and translocation efficiency and fidelity [30]. Hu and co-workers verified the reliability of this technique by successfully unraveling the sequence selectivity of two commonly used drugs, Hg^2+^ and daunomycin [31]. Remarkably, the disparity of the specific binding between two optical isomers of D, L-tetrahydropalmatine (THP) and DNA was found. Through FIRMS, subsequent researchers demonstrated that the insulin-like growth factor type I (IGF-1) selectively binds more tightly to G-quadruplex structures with parallel topology, instead of single/double-stranded DNAs [32]. The quadruplex-interactive ligands were found to cause diminishment to the interaction between IGF-1 and G-quadruplex structures. The information of the IGF-1/quadruplex pathway promoted the research in cancer therapeutics. Furthermore, the integrated use of FIRMS and a general confocal microscope led to the discovery of the mechanical property change concerning intractable tumor cells after the treatment of photodynamic therapy. The results show that the destruction of the extracellular matrix by photodynamic therapy markedly reduced the adhesion force of drug-resistant cells, which restrained the tumor growth and metastasis [33].

### 2.3. Ultrasound Radiation Force

The main advantage of the ultrasound radiation force over the two kinds of force previously mentioned is the integration of the arrays of advanced sensors. The ultrasonic transducer is small enough to be placed inside the atomic magnetometer, which avoids artificial errors during the manual transfer of samples between the force generator and signal detector. The force produced by precisely adjusted ultrasound radiation is capable of resolving the binding differences, such as antibody subclasses and single-base-pair DNA duplexes (Figure 4c) [34]. Subsequently, as the super-resolution force spectroscopy (SURFS), which is based on the integration of the atomic magnetometry and ultrasound techniques, has improved force resolution from 2 to 0.5 pN, the single hydrogen bond between drugs and DNA can be measured [35]. The increase in DNA binding force attributed to drug binding was observed to associate with the enthalpy change, thus providing an alternative physical parameter for optimizing chemotherapeutic drugs. Additionally, the SURFS provided insights into ribosome research with sub-nucleotide (nt) resolution. Xu’s team unveiled that the ribosome spontaneously translocated forward by approximately 0.5 nt before translating the new codon [36]. Recent results found a new position of ribosome during frameshifting [37,38].

### 2.4. Fluid Shear Force

The fluid shear force generated through a parallel-plate flow chamber is tiny and can be integrated into the atomic magnetometer, realizing auto-injection detection. The fluid shear force can be calculated by the following equation:*F* = 6π*ηrv*
where *η* is the viscosity of solution, *r* is the radius of the magnetic probes, and *v* is the fluid velocity. The fluid shear could have merit over SURFS in controlling the force more precisely. Feng and co-workers used a parallel-plate flow chamber to apply fluid shear force and analyzed the through-bond effects on the bivalent binding of the thrombin–aptamer complex (Figure 4d) [39]. The satisfactory results were expected to design high-performance ligands efficiently.

## 3. Magnetic Probes

In addition to the external mechanical forces, another important factor is magnetic probes. In a bid to meet the requirements of the remanent magnetic signal and FIRMS spectrum precision, molecular and cellular system usually use two different types of magnetic probes, microbeads M280 and nanoparticles.

### 3.1. M280

M280 with iron content is a commercial product. Take, for example, Invitrogen company’s Dynabeads^®^ M-280 Streptavidin. They are superparamagnetic materials with a monodisperse diameter of 2.8 μm and size distribution CV < 3%. The monodispersed size ensures a uniform force during external force application. Additionally, the feature of M280 include a hydrophobic bead surface, low charge, and albumin from bovine serum as a blocking protein, which greatly reduce nonspecific adsorption. The surface with covalently coupled streptavidin allows the beads to rapidly bind to the desired target. The molecular interactions detected via magnetic probe M280 include antigen/antibody, DNA/RNA, and aptamer/protein.

### 3.2. Magnetic Nanoparticles (MNPs)

Since MNPs possess significant magnetism, the biocompatible and monodisperse MNPs are optional FIRMS probes for the cell system. The thermal decomposition is one of the most commonly used methods to synthesize monodisperse MNPs due to the advantages of simple operation and chemical inertness. In this route, the non-magnetic organometallic precursor is heated in a container with organic solvents and surfactants [40]. Generally, acetylacetonate irons function as non-magnetic precursor, and the surfactants are normally fatty acids [41]. The optimum range of temperature for this reaction is between 100 °C and 350 °C. Through changing the reaction time and temperature, it is easy to control the MNP size between 4 and 30 nm. When applied to the biological system, MNPs synthesized in an organic phase require improving their surface hydrophilicity through hydrophilic ligand modification or coating with a hydrophilic material.

Yao’s group labeled cells with MNPs in two ways. First, they added MNPs into the bacterial suspension and mixed to disperse them uniformly, helping the magnetic nanoprobes absorb on the surface of the bacteria [42]. Alternatively, they achieved intracellular magnetic labeling [43]. The labeling process is shown in Figure 5. This was performed by incubating MNPs and cells together in serum-free culture medium, where MNPs crossed cell membranes via endocytosis to the interior of the cells. The truth that cell viability was unaffected after MNP labeling brings hope to in vivo experiments. For example, to examine if matrix stiffness induces a pericyte–fibroblast transition, the adhesion force of pericytes on the substrates with different stiffness was measured by MNP-based FIRMS [44].

## 4. Prospects and Outlook

In this review, we presented the main features and applications of FIRMS offered by various external mechanical forces (Table 1) and magnetic probes. The integration of a force generator and atomic magnetometer eliminates the manual error. Through the improvement of detection efficiency and accuracy, the research system was extended to a surface molecule function, through-bond effects, and drugs–DNA interaction. The ingenious DNA reference ruler realizes the non-invasive measurement of biological processes. MNP-based FIRMS are applied to probe the cell adhesion force and offer an alternate means for understanding cell mechanics. These might be of high value in practical applications, including the high-performance ligand design, high-throughput drug screening, clinical diagnosis, and treatment. Alongside these developments, new improvements for high sensitivity, reliability, or handheld miniaturized assays will enable the continuing development of novel FIRMS techniques, which could have far-reaching applications in biomedical research and clinical diagnostics.

## Figures and Tables

**Figure 1 molecules-27-02072-f001:**
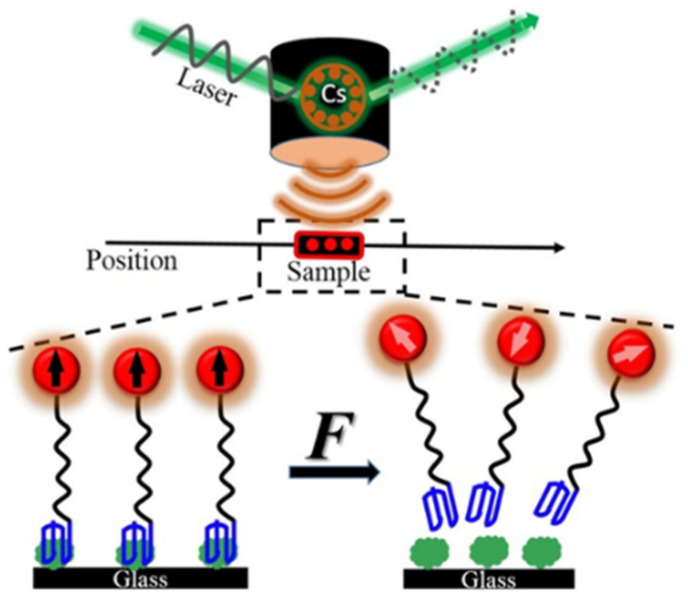
Schematic principle of FIRMS.

**Figure 2 molecules-27-02072-f002:**
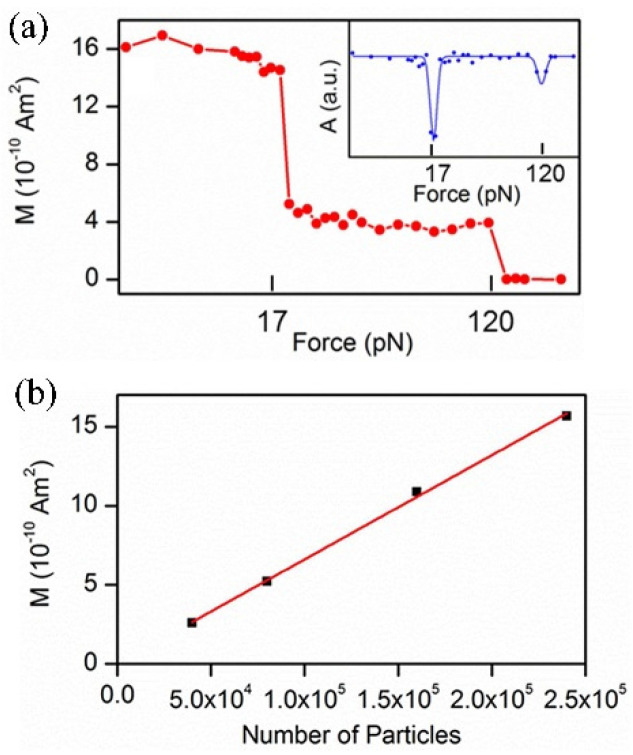
(**a**) FIRMS measurement of noncovalent antibody–antigen bonds, showing the relationship between the magnetization signal and applied force, and the (**b**) calibration for the number of magnetically labeled molecules, showing the linear correlation between the magnetization and the number of various magnetic probes [25]. Reproduced with permission from American Chemical Society.

**Figure 3 molecules-27-02072-f003:**
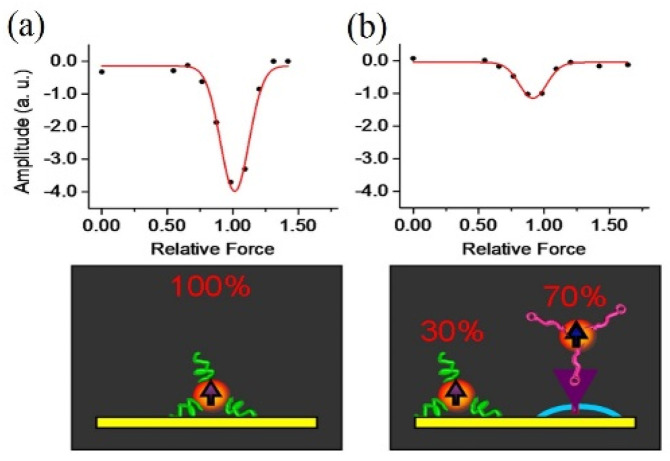
FIRMS spectra showing the magnetization differential as a function of disturbing force for (**a**) the blank and (**b**) cell-binding experiment [26]. Reproduced with permission from Wiley.

**Figure 4 molecules-27-02072-f004:**
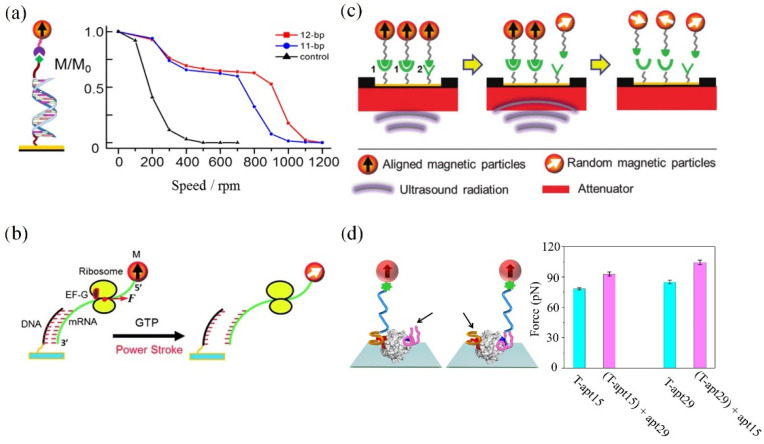
(**a**) FIRMS spectra of the interaction of DNA oligomers [28]. Reproduced with permission from the American Chemical Society. (**b**) Schematic of measuring the EF-G power stroke with internal force references [29]. Reproduced with permission from Wiley. (**c**) Schematic of the ultrasound radiation force-based FIRMS technique [34]. Reproduced with permission from the Royal Society of Chemistry. (**d**) Differential binding force directly proving the through-bond effects in monovalent binding complexes [39]. Reproduced with permission from Wiley.

**Figure 5 molecules-27-02072-f005:**
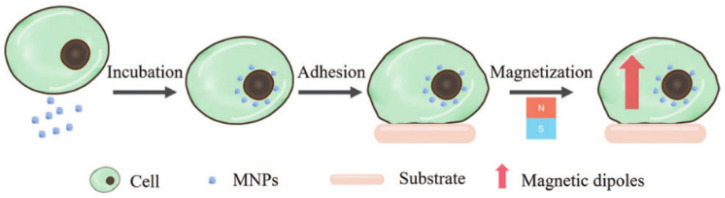
Schematic of intracellular magnetic labeling [43]. Reproduced with permission from the Royal Society of Chemistry.

**Table 1 molecules-27-02072-t001:** Comparison of the external mechanical force in FIRMS.

	Shaking	Centrifugation	Ultrasound Radiation	Fluid Shear
Force generator	Vortex mixer	Centrifuge	Ultrasonic transducer	Parallel-plate flow chamber
Force resolution level (pN)	Low	Medium (~2.0)	Medium (~0.5)	High (~0.5)
Typical applications	Antibody–antigen bonds	Antibody–antigen bonds; multivalent interactions; DNA/RNA system; protein–protein bonds on cell surface; cell adhesion	Ribosomal frameshifting and motion; drug–DNA system;	Protein–aptamer complex
Features	Non-invasive	High throughput	Clinical	Integration

## Data Availability

Not applicable.

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
