# Peer review of "Advances in the Biological Application of Force-Induced Remnant Magnetization Spectroscopy"

_molecules, 2022, doi:10.3390/molecules27072072_

Round 1
Reviewer 1 Report
This is a review of the FIRMS technique by one of the major players in the field. It focuses on details of (1) the method of applying force to the system and (2) the magnetic probes to use, and also includes examples of successful applications of the technique. The paper is well organized and provides good coverage of the topic.
A few improvements can be made to the text:
- "Biology" in the title and on line 82 should be "biological".
- The caption of Figure 2 could be expanded so that it is not necessary to go back to the reference to understand what the figure is showing.
- Figure 3 is also hard to understand - there is no scale on the X-axis, and no explanation of how the plot leads to a figure of 30%/70% for the different types of binding.
- I don't understand the meaning of the phrase "assisting in refinement of drugs with physical quantity" in lines 161-162.
- A number of minor suggested edits, in the attached document, would improve readability.

Reviewer 2 Report
This review provides an overview of the application of Force-induced Remnant Magnetization Spectroscopy to measure non-covalent forces between biomolecules. I found this topic is very interesting although still little known so this work can be useful in spreading the knowledge about it. I consider that the review is relevant and well-structured.
Interesting examples of measurements obtained with the technique are reported and nicely summarized in text, with the support of 5 Figures and 1 Table. In my opinion, just some minor issues could be taken into account before consideration for publication in molecules:
a) some sentences are not clear; I suggest rephrasing them for improving their comprehensibility:
Lines 74-75, p3: […] the labeled samples gradually detach from the immobile undergoing Brownian relaxation.
Lines 82-84, p3: […] intending as a favor in selecting the suitable mechanical force and probe to detect non-covalent interactions.
Lines 139-141, p4: Through FIRMS, subsequent researchers demonstrated that the insulin-like growth factor type I (IGF-1) higher selectively binds to G-quadruplex structures with parallel topology, instead of single/double stranded DNAs [31].
Lines 161-162, p4: […] which assisting in refinement of drugs with physical quantity.
Some minor language issues should be addressed along with the text even if they do not hinder text comprehension.
b) Line 137, p4: Hg2+ → Hg2+
c) Line 158, p4: I think the “super-resolution force spectroscopy (SURFS)” is not clearly introduced in the text. It appears to me as a different technique to FIRMS, while according to ref 34 it seems to refer to the combination of atomic magnetometry and ultrasound. Could you please clarify?
d) Are there any significant applications of microbeads M280?
e) It seems to me that some papers about FIRMS were not included in the review. For example, Yin H, et al doi: 10.1002/cbic.201900276 and doi: 10.3791/59918. which also seems connected to the references 29 and 36 respectively. Could you please explain the reasons for the exclusions?
Reviewer 3 Report
This manuscript offers a short review on the technique of FIRMS (force-induced remnant magnetization spectroscopy) where an external magnetic force drives dissociation in molecular probes attached to the samples. The variation of the magnetic signal as a function of the applied force provides information on the binding forces involved.
The ability to study the forces and motions associated with molecules in biological systems, from cells to proteins, has been revolutionized over the last 20 years by the development of techniques that permit measurements of force and displacement. It is therefore good news that Molecules journal will publish a Special Issue on this topic, and this manuscript is well suited for it.
This review is written in an exhaustive manner and is recommended for publication, although its length does not allow the authors to go deeper into this important topic.
The introduction is sufficiently complete and appropriate. After the introduction, various techniques to apply the required force are illustrated: shaking, centrifugation, ultrasound radiation, and fluid shear forces. The list of techniques reviews is satisfactorily exhaustive, and their specific sensitivities are also reported. After that, the use of two different magnetic probes is also illustrated: microbeads M280 and magnetic nanoparticles.
There are no major problems in the text. Moreover, the figures used in the manuscript are well done. The only important comment is in the decidedly poor resolution of some of them. I refer precisely to figure 3 and in particular to figure 4. I have taken care to check the pdf file provided and have noticed problems of definition and sharpness; furthermore there are some strange halos around the curves of figure 4a) and 4d).
I have checked the publications from which the figures were drawn, and there they are clearly of superior quality. It would be very useful for the manuscript if the authors could show these figures in their original quality.
Furthermore, Figure 3 does not show the label of the horizontal axis ("force" or "relative force") which was instead present in Fig. 3 of the article doi: 10.1002/anie.201007297. The authors should complete it.
The conclusions are well drawn. The authors' use of a table to summarize the different techniques for applying force is commendable, because it offers clarity in their comparison.
On the other hand, I am a bit disappointed by the last sentence, in which they wish "new improvements" for the development of this methodology. I really wish the authors could illustrate what the improvements could be, because it would add interest to this publication.
In the bibliography, there is a large part of the publications of the authors themselves; on the other hand, this is understandable, as this technique is quite "niche".
I have to correct the use made by the text of the reference [28] on line 130. The authors write "Lashan et al." but they should write "De Silva et al.".
